# In Vitro and In Vivo Antibacterial Activity, Toxicity and Resistance Analysis of Pleuromutilin Derivative Z33 against Methicillin-Resistant *Staphylococcus aureus*

**DOI:** 10.3390/molecules27154939

**Published:** 2022-08-03

**Authors:** Yuhan Hu, Fang Chen, Kexin Zhou, Zhe Zhang, Fei Li, Jianfeng Zhang, Youzhi Tang, Zhen Jin

**Affiliations:** 1Guangdong Provincial Key Laboratory of Veterinary Pharmaceutics Development and Safety Evaluation, College of Veterinary Medicine, South China Agricultural University, Guangzhou 510642, China; hyh45020@163.com (Y.H.); 15606993269@163.com (F.C.); zkx2805351587@163.com (K.Z.); z6hang@126.com (Z.Z.); lifeifighting@163.com (F.L.); zjf1339550533@163.com (J.Z.); 2Guangdong Laboratory for Lingnan Modern Agriculture, Guangzhou 510642, China

**Keywords:** pleuromutilin derivative, pharmacodynamic, acute toxicity, CYP450, MRSA

## Abstract

The novel pleuromutilin derivative, which showed excellent in vitro antibacterial activity against MRSA, 22-(2-(2-(4-((4-(4-nitrophenyl)piperazin-1-yl)methyl)-1H-1,2,3-triazol-1-yl)acetamido)phenyl)thioacety-l-yl-22-deoxypleuromutilin (**Z33**), was synthesized and characterized in our previous work. In this study, the preliminary pharmacodynamics and safety of **Z33** were further evaluated. In in vitro antibacterial activity assays, **Z33** was found to be a potent bactericidal antibiotic against MRSA that induced dose-dependent growth inhibition and long-term post-antibiotic effect (PAE). The drug-resistance test demonstrated that **Z33** possessed a narrow mutant selection window and lower propensities to select resistance than that of tiamulin. Cytochrome P450 (CYP450) inhibition assay determined that the inhibitory effect of **Z33** was similar to that of tiamulin against the activity of CYP3A4, and was lower than that of tiamulin on the activity of CYP2E1. Toxicity determination showed that both **Z33** and tiamulin displayed low cytotoxicity of RAW264.7 cells. Furthermore, **Z33** was found to be a high-security compound with a 50% lethal dose (LD_50_) above 5000 mg/kg in the acute oral toxicity test in mice. In an in vivo antibacterial activity test, **Z33** displayed better therapeutic effectiveness than tiamulin in the neutropenic mouse thigh infection model. In summary, **Z33** was worthy of further development as a highly effective and safe antibiotic agent against MRSA infection.

## 1. Introduction

*Staphylococcus aureus* (*S. aureus*) is a major Gram-positive pathogen associated with a range of human diseases from superficial infections of the skin to life-threatening systemic infections such as sepsis, endocarditis, pleuropneumonia, and osteoarthritis [1]. Methicillin-resistant *S. aureus* (MRSA) refers to a type of *Staphylococcus* that can resist most β-lactam antibiotics including penicillin, oxacillin, and amoxicillin [2]. Moreover, MRSA can also be resistant to a variety of other classes of antibiotics in the clinic [3]. Thus, MRSA is usually regarded as a common representative of multidrug-resistant bacteria. Although MRSA was first reported in the 1960s, it had become a worldwide health problem for its wide spread in both communities and hospital settings [4]. The patients infected with MRSA usually suffer more effects including extra cost, extended hospital stays, higher morbidity and mortality than patients infected with methicillin-susceptible *S. aureus* (MSSA) [5]. Therefore, the development of novel antibacterial agents which have unique antimicrobial mechanisms is urgent to combat the emerging challenges of MRSA infections.

Pleuromutilin (Figure 1a), a natural antibiotic with a fused 5-6-8 tricyclic diterpenoid structure, was first isolated from *Pleurotus mutiliz* and *Pleurotus passeckeranius* in 1951 [6]. The pleuromutilin derivatives could inhibit the translation process of bacterial protein by selectively acting on the bacterial 50S ribosomal subunit [7]. The distinct mechanism of action implies a low probability of cross-resistance between pleuromutilin and other antibiotics. Modifications on the C14 side chain of pleuromutilin led to the successful development of tiamulin (Figure 1b), valnemulin (Figure 1c), retapamulin (Figure 1d), and lefamulin (Figure 1e). Among them, lefamulin, the first pleuromutilin antibiotic for human systemic therapy, was approved by Food and Drug Administration (FDA) to treat community-acquired bacterial pneumonia in 2019. Azamulin (Figure 1f), the first pleuromutilin derivative developed for human use, was impeded in phase I clinical trials because of its potent inhibition effect on CYP3A4 [8]. Thus, the development of pleuromutilin derivatives also needed to consider its CYP450 inhibition effect [9]. The development of novel pleuromutilin-type antibiotics, which are highly efficiency and have low toxicity, is an effective strategy against MRSA [10].

In our previous work, a novel pleuromutilin analogue **Z33**, 22-(2-(2-(4-((4-(4-nitrophenyl)piperazin-1-yl)methyl)-1H-1,2,3-triazol-1-yl)acetamido)phenyl)thioacety-l-yl-22-deoxypleuromutilin (Figure 2), was successfully synthesized and characterized [11]. It displayed promising antibacterial activity against MRSA in vitro. Therefore, a series of preliminary pharmacological and toxicology effects of **Z33** were further investigated in this research, including in vitro and in vivo antibacterial activity, acute oral toxicity, and the inhibition effect against CYP450. Meanwhile, the drug-resistant tendency suppression potential of **Z33** and tiamulin was evaluated and compared.

## 2. Results

### 2.1. In Vitro Antibacterial Activity

The results of minimum inhibitory concentrations (MICs) and minimum bactericidal concentrations (MBCs) for four S. aureus strains are summarized in Table 1. Tiamulin and valnemulin were used as reference drugs. The MIC values of **Z33** against four *S. aureus* strains ranged from 0.125 to 0.25 μg/mL. The antibacterial activity of **Z33** was comparable to those of valnemulin and was four-fold greater than that of tiamulin. The ratio of MBC to MIC of **Z33** ranged from 1 to 2. The results demonstrated that **Z33** possessed a potent in vitro bactericidal ability against *S. aureus* (MBC/MIC < 4).

The time–kill assay against MRSA is presented in Figure 3. **Z33** showed inhibition of bacterial growth at 8 × MIC (1 μg/mL) and completely bacterial killing at concentrations ≥ 16 × MIC (2 μg/mL) over 24 h of exposure in the ATCC 43300. The antibacterial activity was positively correlated with the increase in drug concentration. The results indicated that **Z33** exerted antibacterial activity in a concentration-dependent manner.

To investigate the in vitro antibacterial pharmacodynamic activity of **Z33**, the PAEs of **Z33** are evaluated. The results of PAEs are presented in Table 2. **Z33** exhibited PAEs of 2.24 h and 3.89 h at 2 × MIC and 4 × MIC after exposure for 1 h, respectively. The same concentrations of **Z33** exhibited similar PAEs of 2.18 h and 3.93 h after exposure for 2 h. As shown in Figure 4, the PAE of **Z33** against MRSA was prolonged with the increased drug concentration, but not related to the duration of exposure to **Z33**.

### 2.2. In Vitro Drug-Resistance Test

To evaluate the resistant tendency suppression potential of **Z33** and tiamulin, the mutant prevention concentrations (MPCs) of **Z33** and tiamulin against four *S. aureus* strains were measured. As shown in Table 3, the MPCs of **Z33** to four *S. aureus* strains were 0.5 μg/mL, and the MPCs of tiamulin ranged from 1 μg/mL to 2 μg/mL. **Z33** exhibited a low mutation frequency similar to that of tiamulin.

The propensity for the development of bacterial resistance of **Z33** and tiamulin was presented in Figure 5. Four *S. aureus* strains were daily passaged in the exposure of **Z33** and tiamulin for 13 days. In the present study, resistance was defined as a ≥ 4–fold increase in MIC relative to the initial MIC [12]. It was observed that none of *S. aureus* strains tested produced clones with MICs against **Z33** that increased more than 4-fold in 7 passages. In comparison, all *S. aureus* strains produced tiamulin-resistant clones after 7 passages with MICs ranging from 2 to 8 μg/mL. However, approximately 16- and 32-fold increases in MICs were observed for **Z33** and tiamulin after 13 passages. The results indicated that *S. aureus* did not develop resistance against **Z33** within 7 days.

### 2.3. Inhibitory Effects on Cytochrome P450

It has been reported that some pleuromutilin derivatives displayed a potent inhibition effect of CYP450, especially azamulin [8]. The inhibition potential of **Z33** on two major CYP450 enzymes was evaluated using specific CYP probe substrates. Tiamulin was also evaluated for comparison. The CYP450 inhibition was analysed by determining IC_50_ values, and the results are presented in Figure 6.

As shown in Figure 6, both **Z33** and tiamulin exhibited intermediate to high inhibition of CYP3A4 (IC_50_ < 2 μM), with IC_50_ values of 1.575 and 1.598 μM, respectively. However, **Z33** showed much lower inhibition of CYP2E1 (IC_50_ > 10 μM) compared with tiamulin with high inhibition of CYP2E1 (IC_50_ < 0.25 μM) [9].

### 2.4. Cytotoxicity

The cytotoxic effects of **Z33** and tiamulin were evaluated using the MTT assay on RAW264.7 cells. No significant cytotoxicity was observed for **Z33** and tiamulin at concentrations less than 5 μg/mL during the experiment (Figure 7). Up to 20 μg/mL, **Z33** and tiamulin did not result in a <50% reduction in cell activity. Therefore, the IC_50_ for drugs could not be determined. The result indicated that **Z33** had a low inhibition effect on RAW264.7.

### 2.5. Acute Oral Toxicity Study

The results of the acute oral toxicity study are presented in Figure 8. The oral administration of **Z33** at a single-dose of 5000 mg/kg did not induce any remarkable alterations in the behaviour pattern of ICR mice. None of the mice died up to 14 days post-treatment.

### 2.6. In Vivo Antibacterial Activity

The neutropenic mice thigh infection model was constructed to investigate the in vivo antibacterial activity of tiamulin and **Z33** against MRSA. The results are shown in Figure 9. After treatment for 24 h, the logarithmic values of MRSA inoculum in the thigh muscle of the **Z33**-treated group, tiamulin-treated group, and untreated group were 7.065 ± 0.070 log_10_ CFU/g, 7.655 ± 0.067 log_10_ CFU/g, 8.423 ± 0.160 log_10_ CFU/g, respectively. The bacterial load in the thigh muscle was reduced by approximately 1.358 log_10_ CFU/g in the **Z33**-treated group compared to 0.771 log_10_ CFU/g in the tiamulin-treated group. The results indicated that **Z33** at the dose of 20 mg/kg was significantly (*p* < 0.001) more effective against MRSA than tiamulin in vivo.

## 3. Discussion

Human health was continually threatened due to the rise and widespread prevalence of MRSA [4]. Finding novel antibiotics to treat MRSA infections is crucial. The novel pleuromutilin, **Z33**, was shown in this study to be promising antibacterial activity against *S. aureus*, including MRSA and MSSA. The results were similar to the previous reports [11]. Meanwhile, **Z33** might possess a potent bactericidal effect, with MBC/MIC values of 1 to 2 for four *S. aureus* strains. To evaluate the antibacterial efficacy of **Z33** as a function of both time and concentration, a time–kill test was conducted. As shown in Figure 3, **Z33** had a concentration-dependent bactericidal effect against MRSA. **Z33** was completely bactericidal at 2 μg/mL against MRSA ATCC 43300 after 24 h. Because of its excellent in vitro activity against MRSA, **Z33** possessed the appealing potential for clinical use.

The post-antibiotic effect is also an important in vitro pharmacodynamic parameter of novel antibiotics [13]. In present study, **Z33** was found to be a marked concentration-dependent antibacterial activity against MRSA. Our previous study investigated that the PAEs of tiamulin against MRSA ATCC 43300 ranged from 1.65 to 2.04 h after exposure for 2 h [11]. The PAE of **Z33** was 0.5~2 h longer than that of tiamulin under the same experimental conditions. The longer PAE permitted drug levels to fall below the MIC for longer periods of time without loss of efficacy [14]. It is indicated that **Z33** might support longer dosing intervals for clinical treatment of MRSA infection compared with tiamulin.

The development of novel antibiotics with a low propensity to select resistance is one of the effective ways to overcome resistance in bacterial pathogens. We initially evaluated the resistance potential for **Z33** and tiamulin through a mutant prevention study. none of resistant mutants was isolated above a concentration of 4 × MIC (0.5 μg/mL). This suggested that **Z33** had a narrow mutant selection window (MPC/MIC = 2~4) and that might not be susceptible to selectively enriching for resistant mutant subpopulations. To explore whether resistance might arise following prolonged exposure at sub-MIC concentrations, multistep resistance selection was conducted. As shown in Figure 5, resistance to **Z33** developed in four *S. aureus* strains following 8 passages and the most resistant strain showed a 32-fold decrease in **Z33** susceptibility after 13 passages. In comparison, resistance to tiamulin arose earlier following 3 passages. Therefore, compared with tiamulin, **Z33** possessed lower resistance propensities to select resistance. It is indicated that *S. aureus* would be unlikely to rapidly develop resistance against **Z33**.

Cytochrome P450 enzymes (CYP) are involved in the metabolism of more than 90% of drugs in the liver [15]. The CYP3A4 and 2E1 are the most important isoforms among these family members. Previous studies reported that one of the functions of the pleuromutilin derivatives was selective inhibition of CYP3A4 [8,9]. The in vitro incubation assay was used to examine the inhibition potential of pleuromutilin derivatives on CYP450. As shown in Figure 6, **Z33** and tiamulin displayed similar inhibitions of CYP3A4. However, The IC_50_ value of CYP2E1 for **Z33** was at least 50-fold lower than that of tiamulin. **Z33** might show lower hepatotoxicity and less risk of drug–drug interactions.

A desirable feature of any novel antibiotic for clinical use is a low toxicity effect on mammals. Thus, the cytotoxicity of **Z33** was investigated by MTT assay. Tiamulin was also subjected as a reference agent. As shown in Figure 7, both **Z33** and tiamulin displayed IC_50_ of ≥ 20 μg/mL. This result demonstrated that the antibacterial activity of **Z33** is not due to the cytotoxicity effect, but can be attributed to its selective action against bacteria. Furthermore, the acute oral toxicity of ICR mice was investigated. The use of **Z33** at a dose of 5000 mg/kg with 0.5% CMC-Na did not lead to any significant acute toxicological alterations in the mice. In previous work, many pleuromutilin derivatives were assessed as low toxic compounds with the LD_50_ of acute oral toxicity in mice ranging from 2000–3500 mg/kg [16,17]. According to the Globally Harmonized Classification System (GHS) for the classification of chemicals, **Z33** was considered a safe and non-toxic compound.

Since **Z33** exhibited the potent antibacterial activity against MRSA in vitro and high safety in toxicity assays, a neutropenic thigh infection model was conducted to evaluate the efficacy of **Z33** in vivo. In this model, the in vivo antibacterial activity of **Z33** against MRSA ATCC 43300 was quantitatively compared with that of tiamulin. As shown in Figure 9, **Z33** showed potent antibacterial activity against MRSA, resulting in a 1.358 log_10_ reduction in CFUs compared with the control group (*p* < 0.0001) and a 0.65 log_10_ reduction in CFUs compared with the tiamulin-treated group (*p* < 0.001). The efficacy of **Z33** in vivo was concordant with the in vitro antibacterial activity against MRSA in the study. In view of its robust efficacy profile, **Z33** merits further pre-clinical evaluation in in vitro and in vivo development for therapeutic use.

## 4. Materials and Methods

### 4.1. Chemicals

**Z33** (purity: 98%) (22-(2-(2-(4-((4-(4-nitrophenyl)piperazin-1-yl)methyl)-1H-1,2,3-triazol-1-yl)acetamido)phenyl)thioacety-l-yl-22-deoxypleuromutilin) was synthesized in previous work [11]. Tiamulin and valnemulin were purchased from Guangzhou Xiang Bo Biotechnology Co., Ltd. (Guangzhou, China). All chemicals were purchased from Guangzhou Chemical Reagent Factory and TCI (Shanghai) Development Co., Ltd. The medium used for the experiments was purchased from Guangdong Huan Kai Microbial Science and Technology Co. (Guangzhou, China) and formulated according to the instructions. Human liver microsomes were purchased from the Research Institute for Liver Diseases Co., Ltd. (Shanghai, China). 

### 4.2. Microorganisms

MRSA ATCC 43300 was obtained from Guangdong Microbial Strain Conservation Center. *S. aureus* ATCC 29213, *S. aureus* clinical strain AD3 and *S. aureus* clinical strain 144 were stored in our laboratory.

### 4.3. Animals

Six-week-old female specific-pathogen-free (SPF) ICR mice were purchased from the Hunan SJA Laboratory Animal Co., Ltd. (Changsha, China) under license number SCXK2019-0004. All mice were maintained under controlled temperature conditions (20~26 °C), with a constant 12 h light–dark cycle and free access to food and water. All animal experiments were consistent with the ethical principles of animal research and have been approved by the Ethics Committee of Guangdong Medical Laboratory Animal Center.

### 4.4. Evaluation of In Vitro Antibacterial Activity

#### 4.4.1. Determination of MIC and MBC

The MIC values of **Z33** against MRSA ATCC 43300 and *S. aureus* clinical isolates were measured according to the broth microdilution guidelines provided by the Clinical and Laboratory Standards Institute (CLSI) [18]. *S. aureus* ATCC 29213 was used as the quality control strain, and the quality control range was referred to the CLSI. The MIC value is defined as the lowest concentration of the compounds with no visible growth of bacteria. The MIC assay was independently repeated three times.

The MBC values of antibacterial agents were measured after incubation at 37 °C for 24 h. Briefly, 100 µL of bacterial solution from the three wells behind the corresponding well of MIC was absorbed and spread on the Mueller–Hinton Agar (MHA). The MBC refers to the lowest concentration corresponding to the amount of colony-forming unit (CFU) ≤ 5 on the medium after 18~24 h of incubation at 37 °C.

#### 4.4.2. Time–Kill Kinetics Assay

The time–kill kinetics curve of **Z33** was determined according to previous reports [17]. MRSA ATCC 43300 strains were grown in Mueller–Hinton Broth (MHB) at the starting inoculum of 10^6^ CFU/mL. **Z33** and tiamulin were diluted with MHB and added to the bacterial suspension to reach final concentrations from 1 × MIC to 32 × MIC, respectively. MHB without antibiotics was used as a control. At various time points (0, 3, 6, 9, and 24 h), 100 µL of samples were diluted with 0.9% saline and plated on MHA medium. CFU were recorded after 24 h of incubation at 37 °C.

#### 4.4.3. Post-antibiotic Effect Assay

The PAE was measured using the method described by Jin et al. [19]. MRSA ATCC 43300 cells were collected in the logarithmic phase and resuspended in MHB. The bacterial suspension (10^6^ CFU/mL) was inoculated in MHB containing **Z33** (final concentration is 2 × or 4 × MIC) and incubated for 1 or 2 h at 37 °C with shaking (210 rpm). The group without the drug served as a control. After treatment with various concentrations of **Z33**, samples were subjected to one-thousand-fold dilution with MHB to remove the drug and incubated at 37 °C. In total, 100 µL of the samples was serially diluted in 0.9% saline and plated onto MHA at various time points (0, 1, 2, 4, 8 h). MHA plates were incubated at 37 °C for 18~24 h to calculate CFU. The growth kinetic curves of MRSA treated with different drug concentrations were established by plotting log_10_ CFU/mL of bacterial counts versus time. PAE was presented in hours and calculated according to the following equation: PAE = *T*_A_ − *T*_C_, where *T*_A_ and *T*_C_ are the time required for MRSA to rise by 1 log_10_ CFU/mL after drug removal in the treated and untreated cultures, respectively.

### 4.5. Drug-Resistance Test

#### 4.5.1. Determination of MPC

The drug-resistant tendency suppression potential of **Z33** and tiamulin was determined as described with minor modifications [20]. In the test, for each *S. aureus* strain, 4 mL of overnight cultures were inoculated into 100 mL of fresh BHI broth and then incubated for 4.5 h at 37 °C with shaking (210 rpm). Cultures were centrifuged at 5000 r/min for 10 min. The supernatant was discarded and resuspended in the 4 mL of saline to achieve a bacterial density of ~3 × 10^10^ CFU/mL. In total, 100 μL of each culture was spread onto MHA containing various concentrations of **Z33** and tiamulin. The concentration of drugs ranged from 1- to 32-fold MIC. Plates were incubated at 37 °C for a total of 72 h and recorded every 24 h for the appearance of colonies. MPC was defined as the lowest concentration with no growth of colonies on the agar plate. MPC was carried out in triplicate.

#### 4.5.2. Development of Bacterial Resistance

To assess the propensity for the development of bacterial resistance, the ability of **Z33** and tiamulin to induce resistance to four strains were tested, respectively. Serial passages at 37 °C were performed daily in BHI broth containing sub-MIC concentration (1/4 MIC) of **Z33** and tiamulin. The MIC values were daily determined and recorded by the broth microdilution method. A duration of 13 passages was performed in every case.

### 4.6. CYP450 Inhibition Assay

The inhibition effect of **Z33** and tiamulin on the human liver microsomal enzyme was determined as described in the literature with some modifications [9]. Experiments were performed in 96-well plates with a final incubation volume of 100 μL per well. 40 µL of the human liver microsomal enzyme (final concentration is 0.5 g/L), 20 µL of probe substrate (final concentration of testosterone and chlorzoxazone is 20 µmol/L), and 20 µL of test drug were mixed in each well. After preincubation at 37 °C for 5 min, the reactions were started by the addition of 20 μL of NADPH (final concentration is 1 mmol/L). Depending on the substrate, the chlorzoxazone and testosterone groups were incubated at 37 °C for 25 min and 10 min, respectively. The reaction was terminated by adding 100 µL of acetonitrile, and then the supernatant was centrifuged to analyse by LC-MS/MS.

### 4.7. Cytotoxicity Test

To assess the effects of tiamulin and **Z33** on the viability of RAW264.7 macrophages, the MTT assay was performed according to previous reports [21]. Briefly, The RAW264.7 cells inoculated in 96-well plates were treated with different concentrations of tiamulin and **Z33** (from 1.25 to 20 μg/mL) and incubated at 37 °C for 24 h. The plates were cultured at 37 °C for 24 h, followed by incubation with 0.5 mg/mL MTT for 4 h at 37 °C. After discarding the MTT solution, DMSO (150 μL) was added to each well to dissolve all crystals. OD_490nm_ was recorded using EnSight Multimode Plate Reader (PerkinElmer, Walham, MA, USA).

### 4.8. Acute Toxicity in Mice

The acute toxicity of **Z33** was performed in mice according to previous works with some modifications [22]. The ICR mice were randomly divided into two groups (10 animals in each group, half male and half female) after 12 h of fasting. **Z33** was prepared with 0.5% CMC-Na sodium as a suspension at a concentration of 5 g/kg. Mice in the test group were administered with **Z33** suspension by oral administration route, and the control group received an equal dose of 0.5% CMC-Na. All the mice were observed critically once a day for 14 days. The test was repeated twice.

### 4.9. Neutropenic Mouse Thigh Infection Model

The neutropenic mouse thigh infection model was used to determine the in vivo antibacterial activity of **Z33** [23]. Briefly, female mice were rendered neutropenic (neutrophil count ≤ 100/mm^3^) via intraperitoneal injection of cyclophosphamide on day 4 (150 mg/kg) and day 1 (100 mg/kg) before infection. MRSA ATCC 43300 cells were collected in the logarithmic phase and resuspended in 0.9% saline. The thigh infection model was established by injection of 100 μL of MRSA suspension (10^7^ CFU/mL) into the posterior thighs of mice. After 2 h post-infection, mice were intravenously injected with tiamulin and **Z33** at a dose of 20 mg/kg, and the untreated groups were administrated with an equal amount of saline. Mice were euthanized after treatment of 22 h, thigh tissue was aseptically collected, weighed, homogenized, and serially diluted in saline, respectively. CFU counts were determined by the flat colony counting method.

## Figures and Tables

**Figure 1 molecules-27-04939-f001:**
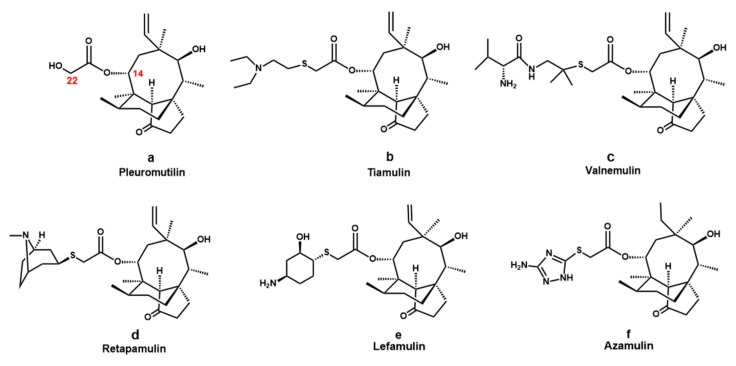
Structure of pleuromutilin (**a**), tiamulin (**b**), valnemulin (**c**), retapamulin (**d**), lefamulin (**e**) and azamulin (**f**).

**Figure 2 molecules-27-04939-f002:**
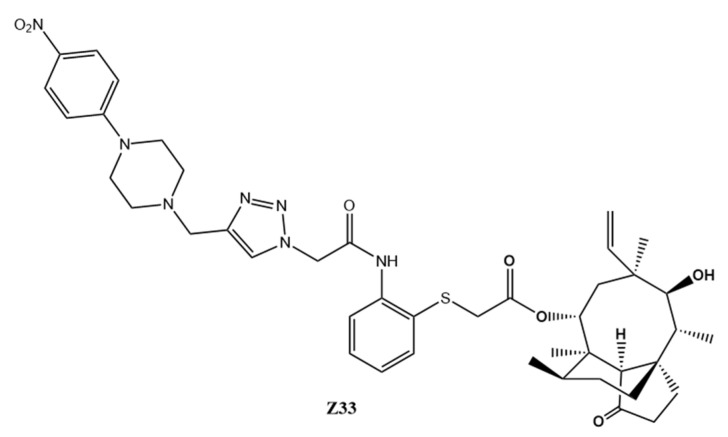
Structure of pleuromutilin derivative **Z33**.

**Figure 3 molecules-27-04939-f003:**
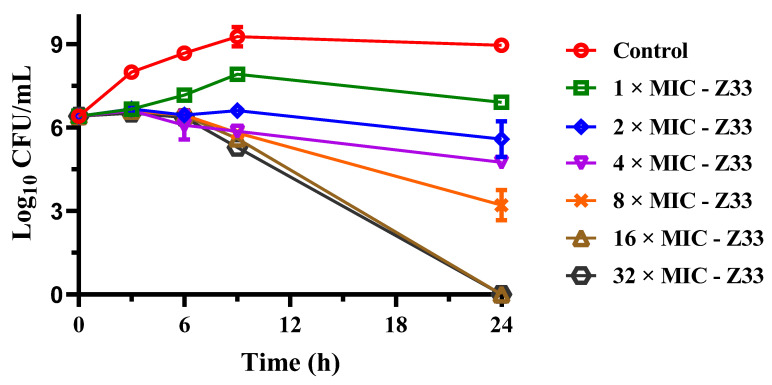
Time–kill curves for MRSA ATCC 43300 with different concentrations of **Z33**.

**Figure 4 molecules-27-04939-f004:**
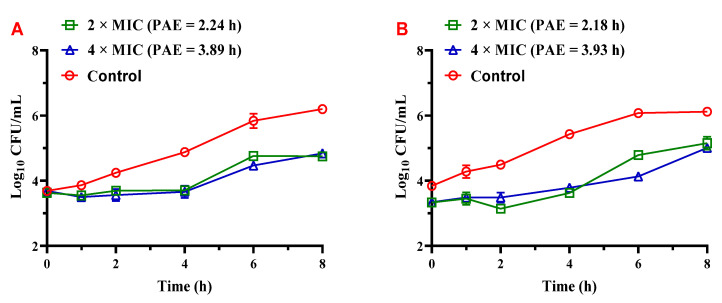
The bacterial growth kinetic curves for MRSA ATCC 43300 exposed to **Z33** for 1 h (**A**) or 2 h (**B**).

**Figure 5 molecules-27-04939-f005:**
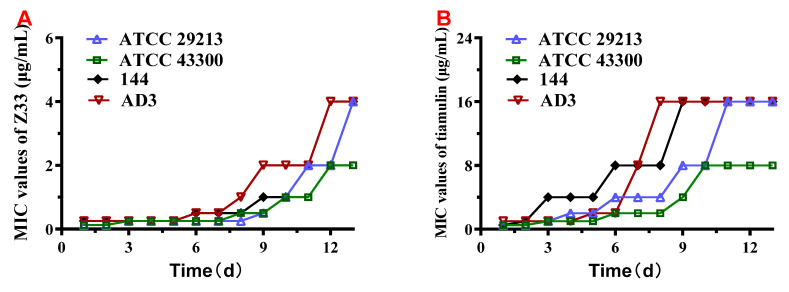
Development of four *S. aureus* strains resistance to **Z33** (**A**) and tiamulin (**B**).

**Figure 6 molecules-27-04939-f006:**
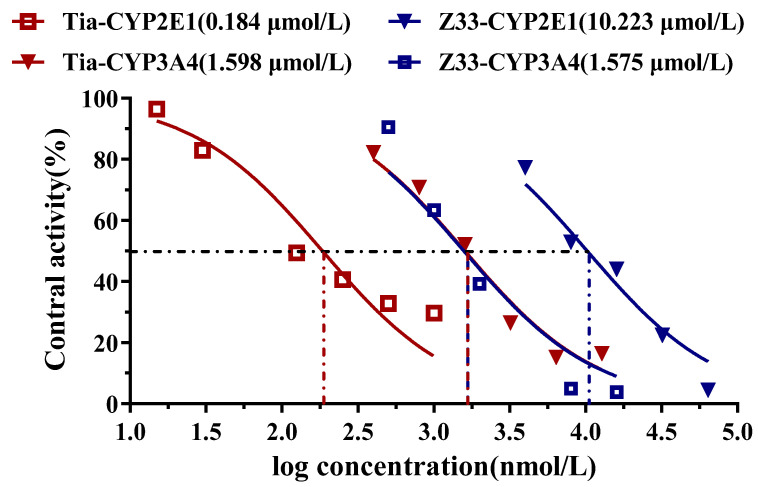
Inhibition curve of **Z33** and tiamulin on CYP3A4 and CYP2E1 (each point in the figure is three parallel averages).

**Figure 7 molecules-27-04939-f007:**
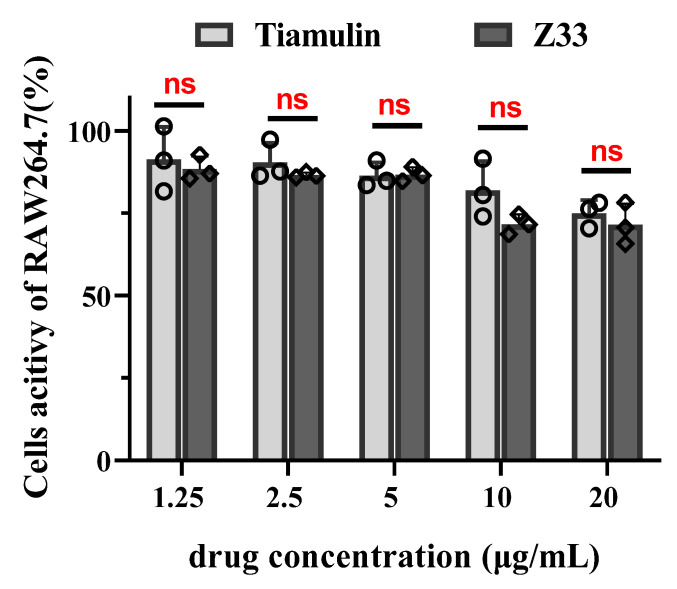
The cytotoxicity assay of **Z33** and tiamulin on RAW264.7 cells. (“ns” stands for no significant difference between both groups, *p* > 0.05).

**Figure 8 molecules-27-04939-f008:**
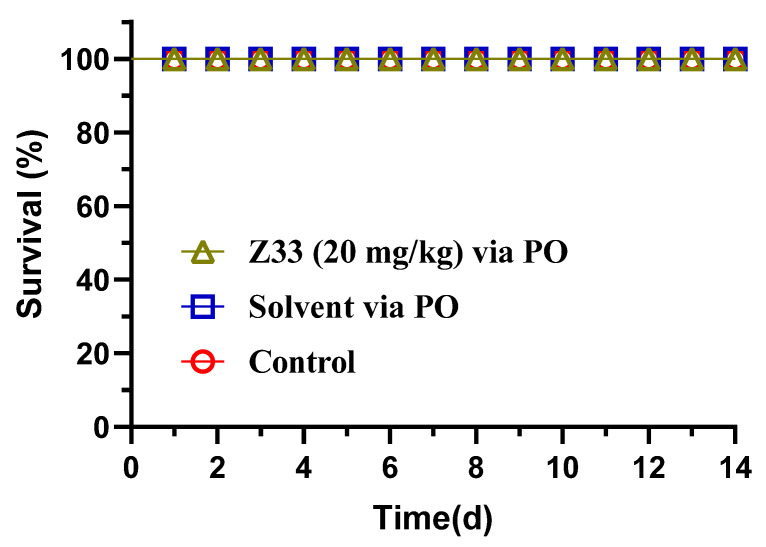
Acute toxicity of **Z33** to mice by oral (PO) routes of administration.

**Figure 9 molecules-27-04939-f009:**
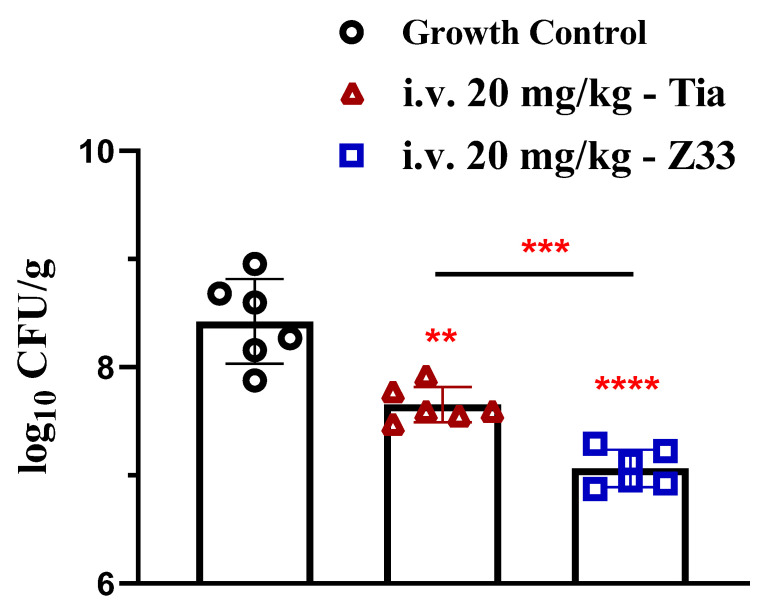
Efficacy of tiamulin (20 mg/kg) and **Z33** (20 mg/kg) against MRSA ATCC 43300 in neutropenic mice thigh infection models. **Z33** vs. growth control, ****; tiamulin vs. growth control, **; **Z33** vs. tiamulin, ***. (** 0.001 < *p* < 0.01; *** 0.0001 < *p* < 0.001; **** *p* < 0.0001).

**Table 1 molecules-27-04939-t001:** MIC and MBC values (μg/mL) of **Z33**, tiamulin, and valnemulin against ATCC 29213, MRSA ATCC 43300, and two clinical strains of *S. aureus* (AD3 and 144).

Compounds	MRSA	ATCC 29213	AD3	144
**Z33**	0.125/0.25	0.125/0.125	0.25/0.25	0.25/0.25
tiamulin	0.5/1	0.5/1	1/2	0.5/2
valnemulin	0.125/0.125	0.125/0.125	0.125/0.125	0.125/0.25

**Table 2 molecules-27-04939-t002:** The PAE values of **Z33** against MRSA ATCC 43300.

Compound	Concentrations	PAE (h)	
		Exposure for 1 h	Exposure for 2 h
**Z33**	2 × MIC	2.24	2.18
	4 × MIC	3.89	3.93

**Table 3 molecules-27-04939-t003:** MPC values of **Z33** and tiamulin against ATCC 29213, MRSA ATCC 43300, and two clinical strains of *S. aureus* (AD3 and 144).

Compounds	MRSA	ATCC 29213	AD3	144
**Z33**	0.5	0.5	0.5	0.5
tiamulin	1	1	2	1

## Data Availability

Not available.

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
