# Peer review of "In Vitro and In Vivo Antibacterial Activity, Toxicity and Resistance Analysis of Pleuromutilin Derivative Z33 against Methicillin-Resistant Staphylococcus aureus"

_molecules, 2022, doi:10.3390/molecules27154939_

Round 1

Reviewer 1 Report

Hu et al describe the in vitro in vivo studies of a lead compound based on their previous studies. The manuscript is suitable for publication after correcting some types and many mistakes. I have highlighted a few of them in the attached file. 

The genus of a biological species should be defined once and then it should be abbreviated. Units should be abbreviated and should be uniform throughout the manuscript.

Author Response

Reviewer #1:

Hu et al describe the in vitro in vivo studies of a lead compound based on their previous studies. The manuscript is suitable for publication after correcting some types and many mistakes. I have highlighted a few of them in the attached file. The genus of a biological species should be defined once and then it should be abbreviated. Units should be abbreviated and should be uniform throughout the manuscript.

Response: Many thanks for the reviewer’s comments. We have carefully checked and improved the English writing in the revised manuscript. The genus of the biological species that first appears in the manuscript has been defined and then abbreviated. Units were abbreviated and uniformly modified throughout the manuscript. All corrections have been listed in Table 1 at the end of the text.

Reviewer 2 Report

Hu et al provide a very straightforward account of preclinical studies for their pleuromutilin derivative Z33.

It seems that one additional opportunity to obtain additional data was missed: it might have been possible to sequence the bacterial gene for 23S rRNA in the resistant strains to investigate the nature of resistance against Z33.

Two edits need to be addressed in a revised version:

1. Reference 20 is inacceptable and must be corrected. Zambia is not a scientific journal. If this is a printed document, cite as a book. At the very least, provide a document identifier.

2. Materials and Methods: More details need to be provided instead of the generic sentence: ‘All other reagents and solvents were purchased commercially and formulated according to the instructions.’ At least provide names of major vendors that supplied reagents.

English language errors (some examples):

1. Introduction, first paragraph: ‘Moreover, MRSA also can resistant to…’

2. Second paragraph: ‘50S ribosomal subunit of bacterial.’

3. ‘also needed to considerate…’

4. page 9: ‘a safe and non-toxicity compound’

Reviewer 3 Report

A study by Hu et al. it is well prepared, the experiments are well designed and the results and conclusions are supported by experimental data. I rate the study as beneficial. I have only minor comments and suggestions for the text: 1/ In many places of the text, the space separating the text and the parenthesis with the quotation is missing 2/ The name of Figure 2 - I suggest adding the name so that it better describes the object - for example "Structure of pleuromutilin analogues Z33". 3/ 2.1 - I recommend omitting the specification of strains in parentheses, redundant. 4/ Fig 4: the explanation of "ns" is missing 5/ 4.4.1 - I recommend deleting the parenthesis with the specification of strains Sa.

Author Response

A study by Hu et al. it is well prepared, the experiments are well designed and the results and conclusions are supported by experimental data. I rate the study as beneficial. I have only minor comments and suggestions for the text:

1/ In many places of the text, the space separating the text and the parenthesis with the quotation is missing.

Response: Many thanks for the reviewer’s comments. The manuscript has been carefully rechecked and revised for formatting and grammar errors.

2/ The name of Figure 2 - I suggest adding the name so that it better describes the object - for example "Structure of pleuromutilin analogues Z33".

Response: Many thanks for the reviewer’s comments. We have changed the note in Figure 2 to "Structure of pleuromutilin derivative Z33". All corrections have been listed in Table 1 at the end of the text.

3/ 2.1 - I recommend omitting the specification of strains in parentheses, redundant.

Response: Many thanks for the editor’s comments. The redundant specifications of strains in 2.1 have been removed in the revised manuscript.

4/ Fig 4: the explanation of "ns" is missing.

Response: Many thanks for the editor’s comments. The explanation of "ns" has been added in Figure 7.

5/ 4.4.1 - I recommend deleting the parenthesis with the specification of strains Sa.

Response: Many thanks for the reviewer’s comments. The statement of strain specifications in parentheses has been deleted.
